# Characterization of *Martelella soudanensis* sp. nov., Isolated from a Mine Sediment

**DOI:** 10.3390/microorganisms9081736

**Published:** 2021-08-14

**Authors:** Jung-Yun Lee, Dong-Hun Lee, Dong-Hun Kim

**Affiliations:** 1Groundwater Research Center, Geologic Environment Division, Korea Institute of Geoscience and Mineral Resources, Daejeon 34132, Korea; 8548@kigam.re.kr; 2Department of Microbiology, Chungbuk National University, Cheongju 28644, Korea; donghun@chungbuk.ac.kr

**Keywords:** *Martelella soudanensis* sp. nov., Soudan Iron Mine sediment, polyphasic taxonomy

## Abstract

Gram-stain-negative, strictly aerobic, non-spore-forming, non-motile, and rod-shaped bacterial strains, designated NC18^T^ and NC20, were isolated from the sediment near-vertical borehole effluent originating 714 m below the subsurface located in the Soudan Iron Mine in Minnesota, USA. The 16S rRNA gene sequence showed that strains NC18^T^ and NC20 grouped with members of the genus *Martelella*, including *M. mediterranea* DSM 17316^T^ and *M. limonii* YC7034^T^. The genome sizes and G + C content of both NC18^T^ and NC20 were 6.1 Mb and 61.8 mol%, respectively. Average nucleotide identity (ANI), the average amino acid identity (AAI), and digital DNA–DNA hybridization (dDDH) values were below the species delineation threshold. Pan-genomic analysis showed that NC18^T^, NC20, *M. mediterranea* DSM 17316^T^, *M. endophytica* YC6887^T^, and *M. lutilitoris* GH2-6^T^ had 8470 pan-genome orthologous groups (POGs) in total. Five *Martelella* strains shared 2258 POG core, which were mainly associated with amino acid transport and metabolism, general function prediction only, carbohydrate transport and metabolism, translation, ribosomal structure and biogenesis, and transcription. The two novel strains had major fatty acids (>5%) including summed feature 8 (C18:1 ω7c and/or C18:1 ω6c), C19:0 cyclo ω8c, C16:0, C18:1 ω7c 11-methyl, C18:0, and summed feature 2 (C12:0 aldehyde and/or iso-C16:1 I and/or C14:0 3-OH). The sole respiratory quinone was uniquinone-10 (Q-10). On the basis of polyphasic taxonomic analyses, strains NC18^T^ and NC20 represent novel species of the genus *Martelella*, for which the name *Martelella soudanensis* sp. nov. is proposed. The type strain is NC18^T^ (=KTCT 82174^T^ = NBRC 114661^T^).

## 1. Introduction

Although deep subsurface environments are characterized by relatively low available organic carbon and elevated pressure, they are the largest habitats for prokaryotes, estimated to contain from 12% to 20% of the total biomass of microorganisms on Earth [1,2]. The deep biosphere contains a variety of functionally active microbial communities, but microbes face challenges such as limited available electron donors or acceptors, pore space, and fracture networks, and competition with other microorganisms for growth [3]. The Soudan Iron Mine is located in northern Minnesota, USA, along the southern edge of the Canadian Shield Transects and the Archaean Animikie ocean basin. The mine reaches a depth of 714 m below the surface, providing access to deep subsurface brines entrained in a massive hematite formation [3,4]. Previous cultivation-based and metagenomics studies reported the presence of diverse microbial communities, including the iron-oxidizing *Marinobacter* and iron-reducing *Desulfuromonas* in a calcium- and sodium-rich brine that reaches salinities as high as 4.2% (*w*/*v*) from these boreholes [3,5,6].

Halophiles and halotolerant bacteria are able to survive even in highly saline environments that are unfavorable for the existence of most life forms [7]. In recent decades, halophilic and halotolerant bacteria have been studied in biotechnological applications [8]. Halophiles have stable enzymes that function in high salinity and are used in industrial processes where high salt concentrations would inhibit enzymatic transformations [9]. Many compounds synthesized by halophiles (e.g., enzymes, polysaccharides, polyhydroxyalkanoates, and biosurfactants) are available for commercial uses [10,11,12,13]. Several halophilic enzymes, including proteases, amylases, lipases, and xylanases, have been characterized in recent years and are used in biofuel production [14], food processing [15], and the biodegradation of organic contaminants [16,17]. These halophilic and halotolerant bacteria are spread over a large number of phylogenetic branches, with most species being grouped in *Proteobacteria* [18].

The genus *Martelella* was originally described in 2005 by Rivas et al., isolated from a subterranean saline lake [19]. This genus belongs to the phylum *Proteobacteria*, the class *Alphaproteobacteria*, the order *Hyphomicrobiales,* and the family *Aurantimonadaceae* (Available online: https://lpsn.dsmz.de/genus/martelella (accessed on 16 July 2021)). Thus far, a total of 8 species were reported in this genus: *Martelella mediterranea* [19], *Martelella endophytica* [20], *Martelella radicis* [21], *Martelella mangrovi* [21], *Martelella suaedae* [22], *Martelella limonii* [22], *Martelella caricis* [23], and *Martelella lutilitoris* [24]. The members of *Martelella* are Gram-stain-negative, aerobic, rod-shaped, non-motile, non-spore-forming, and chemoheterotrophic bacteria. *Martelella* strains were isolated from diverse environments such as a saline lake, the roots or mud plat of halophytes, and the soils of mangrove roots [19,20,21,22,23,24]. Additionally, a halophilic polycyclic aromatic hydrocarbon (PAH)-degrading strain was reported [25]. Therefore, this genus could be of great biotechnological interest, as halotolerant and halophilic microorganisms are well-known for their potential applications in biotechnology.

Here, we report the taxonomic characterization of bacterial strains, NC18^T^ and NC20, isolated from mine sediment by using polyphasic approaches.

## 2. Materials and Methods

### 2.1. Enrichment Culture and Isolation

Strains NC18^T^ and NC20 were isolated from sediment near a descending exploratory borehole (47°49.2′ N, 92°14.5′ W), Soudan Mine Diamond Drill Hole 942, located at the bottom (level 27) of the Soudan Underground Mine State Park in Soudan, Minnesota, USA. The sediment sample was serially diluted with 0.85% NaCl, and suspensions were plated on R2A agar (BD, Franklin Lakes, NJ, USA) supplemented with 2% NaCl (*w*/*v*) and incubated at room temperature for 7 days. Circular, smooth, and white-to-cream-colored colonies of designated strains NC18^T^ and NC20 were isolated and subsequently purified three times. Cultures were maintained on R2A plates supplemented with 2% NaCl at 30 °C, and stocks were preserved in R2A broth with glycerol (20%, *v*/*v*) at −80 °C.

### 2.2. Phenotypic Analysis

Cell morphology and flagellation of strains NC18^T^ and NC20 were observed by transmission electron microscope (CM20, Philips, Amsterdam, The Netherlands) operated at 80 kV with cells grown in a marine agar (MA; BD, Franklin Lakes, NJ, USA) plate for 2 days at 30 °C. Cells were negatively stained using 2% (*w*/*v*) uranyl acetate, air-dried, and had their grids examined. The presence of spores was analyzed by phase-contrast microscopy (ECLIPSE 80i, Nikon, Tokyo, Japan) at a magnification of 1500×, using cells that had been grown for 1 week at 30 °C on MA. Gram staining was determined by using the bioMérieux Gram-staining kit according to the manufacturer’s instructions. The colony color and morphology of strains NC18^T^ and NC20 were investigated on MA plate incubated for 2 days at 30 °C.

Growth at different temperature (4, 10, 12, 15, 20, 25, 30, 35, and 40 °C) and pH (pH 4.0–10.0 at intervals of 1.0 pH units) levels were tested on marine broth 2216 (MB; BD, Franklin Lakes, NJ, USA). pH was adjusted with 0.1 N NaOH or 0.1 N HCl solutions and checked after autoclaving. Salt tolerance was investigated in marine broth supplemented with 0–15% (*w*/*v*, at 1% intervals) NaCl. Growth of strains NC18^T^ and NC20 at different temperatures, pH, and NaCl concentrations were determined by OD_600_ using a spectrophotometric method (Optizen POP, Mechasus, Daejeon, Korea). Growth under anaerobic conditions was determined on MA plates at 30 °C for 10 days in an anaerobic Gaspak jar (OXOID) with Anaero-PACK (Mitsubishi Gas Chemical Co., Tokyo, Japan).

Catalase activity was determined with 3% (*v*/*v*) hydrogen peroxide. Casein hydrolysis was examined on MA supplemented with 1% (*w*/*v*) skim milk. Enzyme activities, acid production from different carbohydrates, and the assimilation of various substrates were determined using API ZYM, API 20E, and API 20NE, respectively, according to the manufacturer’s instructions (bioMérieux, Marcy l’Etoile, France).

### 2.3. Chemotaxonomic Analysis

For the analysis of cellular fatty acid and respiratory quinone, cells of NC18^T^ and NC20 were prepared from cells grown on MA for 2 days at 30 °C. Fatty acid methyl esters were prepared and analyzed according to the standard MIDI protocol (Sherlock Microbial Identification System, version 6.2) and identified by the RTSBA 6.0 database of the Microbial Identification System [26].

Respiratory quinone was extracted by the chloroform-methanol extraction method and analyzed using high-performance liquid chromatography (HPLC) as previously described [27].

### 2.4. Phylogenetic Analysis

Genomic DNA was prepared using an AccuPrep Genomic DNA Extraction Kit (Bioneer, Daejeon, Korea) according to the manufacturer’s instructions. DNA was precipitated using 1 volume of chilled isopropanol and 0.1 volume of 3 M sodium acetate, followed by overnight incubation at −20 °C. DNA pellet was collected by centrifugation at 13,800× *g* for 30 min at 4 °C. The DNA pellet was washed with 70% ethanol, air-dried, and resuspended in nuclease-free water (Qiagen, Germantown, MD, USA).

The 16S rRNA gene was amplified using universal primers 27F and 1492R [28]. The 16S rRNA gene sequences (1482 nt) of strains NC18^T^ and NC20 were identical with the corresponding region of its genomic sequence and were compared with the related sequences from the EzBioCloud server (Available online: www.ezbiocloud.net (accessed on 16 July 2021)) [29]. The 16S rRNA gene sequences were aligned with those of closely related species using the CLUSTAL X software program [30]. Gaps were edited in the BioEdit program [31] using the neighbor-joining, maximum-parsimony, and maximum-likelihood algorithms in MEGA 6.0 software [32]. Bootstrap analysis was performed to determine confidence values of individual branches in the phylogenetic tree with 1000 replications. The 16s rRNA gene sequences of strains NC18^T^ and NC20 were deposited in GenBank/EMBL/DDBJ under accession numbers MT367774 and MT367775, respectively.

Multilocus sequence analysis (MLSA) was also conducted to support phylogenetic analysis using 31 housekeeping gene products: dnaG, frr, infC, nusA, pgk, pyrG, rplA, rplB, rplC, rplD, rplE, rplF, rplK, rplL, rplM, rplN, rplP, rplS, rplT, rpmA, rpoB, rpsB, rpsC, rpsE, rpsI, rpsJ, rpsK, rpsM, rpsS, smpB, and tsf. The available genomic data of the genus *Martelella* were obtained from the GeneBank database (Appendix A). Then, 31 universally conserved proteins were extracted using the AmphoraNet server [33]. A multiple alignment of 31core proteins was built using MUSCLE [34]. The resulting alignment was concatenated using MEGA 6.0 [32], resulting in 9333 amino acid positions. Gaps were then removed using Gblocks version 0.91b [35], resulting in 4988 amino acid positions, covering about 53% of the original alignment. The phylogenetic tree was reconstructed by using the neighbor-joining, maximum-likelihood, and maximum-parsimony algorithms with the JTT matrix-based model [36] and 1000 bootstrap replicates in MEGA 6.0 [32].

### 2.5. Whole-Genome Sequencing, Assembly, and Annotation

Whole-genome sequencing and assembly were carried out using Illumina MiSeq and PacBio RS II platforms at the ChunLab Inc. (Seoul, Korea). Illumina sequencing data were assembled with SPAdes 3.9.0 (Available online: http://cab.spbu.ru/software/spades/ (accessed on 5 December 2020)). MiSeq sequencing data were quality-controlled with Trimmomatic 0.36, and PhiX sequences were removed with BBMap 38.32. PacBio RS II sequencing data were assembled with PacBio SMRT Link 7.0.1 using the HGAP4 protocol (Pacific Biosciences, Menlo Park, CA, USA). Hybrid assembly was performed by Pilon version 1.22 and reassembled using quality-controlled MiSeq data and assembled contigs from Pacbio data. The resulting contigs from the hybrid assembly were circularized using Circlator 1.4.0 (Sanger Institute). Contamination of the whole-genome sequence was checked using ContEst16S [37]. Gene prediction and functional annotation pipeline of whole-genome assemblies used in EzBioCloud database. Protein coding sequences (CDSs) were predicted by Prodigal v.2.6.2 [38]. Genes coding for tRNA were identified using tRNAscan SE 1.3.1 [39]. rRNA operons were predicted by a covariance model search with Rfam 12.0 database [40]. CRISPRs were detected by PilerCR 1.06 [41] and CRT 1.2 [42]. Identified protein coding sequences (CDSs) were classified into clusters of orthologous groups (COGs) on the basis of their roles, with reference to orthologous groups (EggNOG 4.5; Available online: http://eggnogdb.embl.de (accessed on 20 December 2020)) [43]. For more functional annotation, the predicted CDSs were compared with Swissprot [44], KEGG [45], and SEED [46] databases using UBLAST [47]. The genomic sequences of strains NC18^T^ and NC20 were deposited in DDBJ/ENA/GenBank under accession numbers CP054858-CP054860 and CP054861-CP054863, respectively.

### 2.6. Comparative Genomic Analysis

DNA G + C contents of strains NC18^T^ and NC20 were calculated from the whole-genome sequence. Overall genome relatedness index (OGRI) of strains NC18^T^, NC20, and reference strains with available genomic sequences, *M. mediterranea* DSM 17316T (GenBank assembly accession GCA_002043005.1), *M. endophytica* YC6887T (GCA_000960975.1), and *M. lutilitoris* GH2-6T (GCA_005924265.1) were estimated on the basis of average nucleotide identity (ANI) using the ANI calculator employing the OrthoANIu algorithm [48] and digital DNA–DNA hybridizations (dDDH) values using the genome-to-genome distance-calculation (GGDC) method [49]. An online calculator GGDC2.1; available online: http://ggdc.dsmz.de/ggdc.php# (accessed on20 May 2021) was used for calculating the dDDH value with recommended Formula 2 [50]. Identified CDSs were classified into groups on the basis of their roles according to the reference to orthologous groups (EggNOG available online: http://eggnogdb.embl.de (accessed on 20 May 2021) [43]. To calculate the similarity at the orthologous protein level between genomes, given that average amino acid sequences change more slowly than nucleotide sequences do, two-way average amino acid identity (AAI), which is more sensitive over greater evolutionary distances, based on reciprocal best hits, was calculated using the AAI calculator available online: http://enve-omics.ce.gatech.edu/aai (accessed on 20 May 2021) [51]. The Bacterial Pan-Genome Analysis Tool (BPGA) pipeline [52] was used to define core (shared with all five strains), accessory (shared with more than two but not all strains), and unique (strain-specific) pan-genome orthologous groups (POGs) of the five *Martelella* strains. POG clustering was carried out using the USEARCH algorithm with an identity value of 0.5.

## 3. Results and Discussion

### 3.1. Phenotypic Characterization

Cells of strains NC18^T^ and NC20 were short rod-shaped (0.8–0.9 × 1.2–1.3 μm and 0.8–1.1 × 1.2–2.5 μm, respectively; Appendix A), Gram-stain-negative, strictly aerobic, non-motile, non-spore-forming, and oxidase-, and catalase-positive. The colonies of the two strains were cream-colored, smooth, convex, and circular.

Phenotypic examination revealed several common traits between the novel strains and closely related type strains. However, strains NC18^T^ and NC20 could be clearly differentiated from type strains by their ability to maximally grow in a higher NaCl concentration (13%); the ability for L-rhamnose fermentation-oxidation; the presence of urease; the absence of aesculin hydrolysis and acetoin production; and the inability for esterase lipase activity. Strain NC18^T^ could also be differentiated from strain NC20 by its ability to optimally grow at pH 7; ability to maximally grow at a temperature of 40 °C and the presence of β-glucosidase; and N-acetyl-β-glucosaminidase activity. The detailed morphological, physiological, and biochemical characteristics of strains NC18^T^ and NC20 are given in Table 1; the species description is in Section 4.1. Thus, the distinguished phenotypic properties suggest that strains NC18^T^ and NC20 are separated from other species of the genus *Martelella*.

### 3.2. Chemotaxonomic Characterization

Predominant cellular fatty acids of strain NC18^T^ were summed feature 8 (C18:1 ω7c and/or C18:1 ω6c, 37.5%), C19:0 cyclo ω8c (25.8%), C16:0 (13.0%), C18:1 ω7c 11-methyl (6.4%), C18:0 (6.2%), and summed feature 2 (C12:0 ALDE, an unidentified fatty acid with an equivalent chain-length of 10.9525, iso-C16:1 I and/or C14:0 3-OH; 5.9%) (Table 2). The predominant cellular fatty acids of strain NC20 were summed feature 8 (C18:1 ω7c and/or C18:1 ω6c, 35.0%), C19:0 cyclo ω8c (28.2%), C16:0 (13.4%), C18:1 ω7c 11-methyl (6.6%), C18:0 (6.3%), and summed feature 2 (C12:0 ALDE, an unidentified fatty acid with an equivalent chain-length of 10.9525, iso-C16:1 I and/or C14:0 3-OH; 5.6%) (Table 2). The fatty acid profiles of strains NC18^T^ and NC20 had similar patterns to those of members of the genus *Martelella*. However, there were some differences in the respective compositions of some long-chain fatty acids, such as C18:1 ω5c, C19:0 iso, and C20:1 ω7c. The respiratory quinones of strains NC18^T^ and NC20 were Q-10, typical for the genus *Martelella* [1,2,3,4,5,6].

### 3.3. Phylogenetic Characterization

The 16S rRNA gene sequences of strains NC18^T^ and NC20 were 1482 bp, showing 100% similarity to each other. NC18^T^ and NC20 showed the highest sequence similarity values with those of *M. mediterranea* DSM 17316^T^ (99.0%), *M. limonii* YC7034^T^ (98.6%), *M. endophytica* YC6887^T^ (98.1%), *M. mangrovi* BM9-1^T^ (97.9%), *M. lutilitoris* GH2-6^T^ (97.9%), *M. suaedae* YC7033^T^ (97.6%), *M. radicis* BM5-7^T^ (97.6%), and *M. caricis* GH2-8^T^ (97.2%). The 16S rRNA gene sequence similarities indicated that strain *M. mediterranea* DSM 17316^T^ and *M. limonii* YC7034^T^ were the nearest phylogenetic neighbors to the novel isolate. The 16S rRNA gene sequence phylogenetic analysis based on the neighbor-joining (Figure 1), maximum-likelihood (Appendix A), and maximum-parsimony (Appendix A) algorithms indicated that strains NC18^T^ and NC20 formed a lineage within the clade of the genus *Martelella* and separated it from the clade composed of the species *M. mediterranea* and *M. limonii*.

The MLSA tree also agreed with the taxonomic positions of the two strains as shown by the phylogenetic tree based on the 16S rRNA gene (Figure 2, Appendix A).

### 3.4. General Genomic Features

The genomic features of strains NC18^T^ and NC20 are shown in Table 3. The genome sizes of NC18^T^ and NC20 were 6,109,459 and 6,109,677 bp, respectively. The NC18^T^ genome was predicted to have 5849 genes, 5502 protein-encoding genes, 6 rRNAs, and 48 tRNAs. The NC20 genome was predicted to have 5830 genes, 5585 protein-encoding genes, 6 rRNAs, and 48 tRNAs.

The DNA G + C contents of both NC18^T^ and NC20 were 61.8 mol% (Table 1), which are within the known DNA G + C contents of genus *Martelella* (52.8–62.6 mol%). The predicted functional genes based on the COG database of both strains mainly belong to amino acid transport and metabolism (E; 537 and 526 orthologs for NC18^T^ and NC20, respectively), carbohydrate transport and metabolism (G; 534 and 524 orthologs), transcription (K; 500 and 496 orthologs), and inorganic ion transport and metabolism (P; 458 and 451 orthologs), except only a general function prediction (R) (Appendix A).

### 3.5. Comparative Genomic Characterization

ANI, AAI, and dDDH values between NC18^T^ and NC20 were 99.9%, 100%, and 100%, respectively (Table 4 and Table 5). These indicate that the two isolates belonged to a single species. In contrast, ANI values between NC18^T^ and the reference strains of *M. mediterranea* DSM 17316^T^, *M. endophytica* YC6887^T^, and *M. lutilitoris* GH2-6^T^ were 88.1%, 80.2%, and 80.4%, respectively (Table 4). AAI values between NC18^T^ and the reference strains of *M. mediterranea* DSM 17316^T^, *M. endophytica* YC6887^T^, and *M. lutilitoris* GH2-6^T^ were 87.7%, 76.3%, and 77.8%, respectively (Table 5). These ANI and AAI values were lower than the 95% threshold used to identify isolates as belonging to the same bacterial species [53,54]. The dDDH values of strain NC18^T^ and the reference strains of *M. mediterranea* DSM 17316^T^, *M. endophytica* YC6887^T^, and *M. lutilitoris* GH2-6^T^ were 34.9%, 23.9%, and 23.7%, respectively (Table 4), which were below the threshold of 70% for species delineation [53]. Altogether, these results indicate that strain NC18^T^ represents a novel species of the genus *Martelella*.

To gain an in-depth understanding of the intra-species genomic diversity of the *Martelella* species, pan-genome analysis was performed. The pan-genome curve showed that the size of the pan-genome increased with the addition of new genomes (Appendix A). The core genome slowly decreased as the genomes were added one by one. Pan-genomic analysis shows that the two strains and three related species had 8470 POGs: 2258 POG core, 3617 POG accessory, and 2595 POG unique (Figure 3). The five strains contained a certain number of strain-specific genes (POG unique), and the number varied considerably (9–971) depending on each strain, showing that the changing genetic flow led to the generation of strain specificity (Appendix A). Most of the POG core was classified into basic functions: amino acid transport and metabolism (E), general function prediction only (R), carbohydrate transport and metabolism (G), translation, ribosomal structure and biogenesis (J), and transcription (K) (Appendix A), which were related to necessary nutrients obtaining from various environments and maintaining a lifestyle. Function unknown (S) occupied a large proportion, which showed the current lacing in understanding *Martelella* genomes. Most of the POG accessory was also classified as a similar pattern to that of the POG core.

In addition, Kyoto Encyclopedia of Genes and Genomes (KEGG) analysis indicated that the POG unique were mainly involved in carbohydrate metabolism, membrane transport, amino acid metabolism, and lipid metabolism, especially carbohydrate metabolism (Appendix A), which corresponded with the variable ability of carbon source use tested by API (Table 1). These genomic characteristics indicated the diversity of metabolic pathways in different *Martelella* strains. In addition, the concatenated POG core-based phylogenetic tree (Appendix A) showed that the two novel strains were distinct from the three other *Martelella* strains, as clustering with *M. mediterranea* DSM 17316^T^.

This result was consistent with the phylogenetic tree based on 16S rRNA gene sequences of strains NC18^T^ and NC20 with other related taxa. Thus, the distinguished genetic distinctiveness by ANI, AAI, and dDDH values suggest that strain NC18^T^ is separated from other recognized species of the genus *Martelella*. Pan-genomic analysis also showed the distinct patterns of gene content and metabolism pathway within this new *Martelella* species.

## 4. Conclusions

We isolated two *Martelella* strains and performed phenotypic, chemotaxonomic, phylogenetic, and genomic analyses to identify these strains as novel species. Phylogenetic analysis showed that strains NC18^T^ and NC20 grouped with members of the genus *Martelella*, including *M. mediterranea* DSM 17316^T^ and *M. limonii* YC7034^T^. Genetic distinctiveness by ANI, AAI, and dDDH values suggested that strains NC18^T^ and NC20 are separated from other recognized species of the genus *Martelella*. Pan-genomic analysis showed that most POG cores of NC18^T^ and NC20 were relevant to amino acid transport and metabolism in the COG category and carbohydrate metabolism in the KEGG pathway. On the basis of polyphasic taxonomic data, strain NC18^T^ represents the type strain of a novel species of the genus *Martelella*, for which the name *Martelella soudanensis* sp. nov. is proposed.

### 4.1. Description of Martelella Soudanensis sp. nov.

*Martelella soudanensis* (sou.da.nen′sis. N.L. masc./fem. adj. soudanensis, named after Soudan Iron Mine).

Cells are Gram-stain-negative, strictly aerobic, oxidase-positive, catalase-positive, non-spore-forming, non-flagellated rods (0.8–0.9 × 1.2–1.3 μm). Colonies on MA are cream-colored, smooth, convex, circular, and reach 0.5 mm in diameter after incubation for 2 days at 30 °C. Growth occurs at 20–40 °C with an optimum of 30 °C. Growth does not occur at 10 °C or above 45 °C. The pH range for growth is pH 5.0–9.0 (optimum, pH 7.0). Growth occurs in the range of 0%–13% (*w*/*v*) NaCl with an optimum of 1% (*w*/*v*). Positive for alkaline phosphatase, esterase (C4), leucine arylamidase, acid phosphatase, naphthol-AS-BI-phosphohydrolase, α-glucosidase, β-glucosidase, and N-acetyl-β-glucosaminidase. Negative for esterase lipase (C8), lipase (C14), valine arylamidase, cystine arylamidase, trypsin, α-chymotrypsin, α-galactosidase, β-glucuronidase, α-mannosidase, α-fucosidase, lysine decarboxylase, ornithine decarboxylase, acetoin production, H_2_S production, and tryptophane deaminase. Nitrate reduction and urease were observed. Aesculin hydrolysis, glucose fermentation, gelatin hydrolysis, indole production, arginine dihydrolase, caseinase are not observed. Potassium gluconate, malate, D-maltose, adipate, caprate, D-mannose, N-acetyl-D-glucosamine, phenylacetate, D-glucose, D-mannitol, L-arabinose, and trisodium citrate are not used as carbon and energy sources. Positive for acid production from L-rhamnose and L-arabinose, but negative for acid production from amygdalin, inositol, D-melibiose, D-sorbitol, D-sucrose, D-glucose, and D-mannitol. The major respiratory quinone is Q-10. Predominant fatty acids are summed feature 8 (C18:1 ω7c and/or C18:1 ω6c), C19:0 cyclo ω8c, C16:0, C18:1 ω7c 11-methyl, C18:0, and summed feature 2 (C12:0 aldehyde and/or iso-C16:1 I and/or C14:0 3-OH). The type strain, NC18^T^ (=KCTC 82174^T^ = NBRC 114661^T^), was isolated from mine sediment in Soudan, Minnesota, USA. The genome size of the type strain is 6.10 Mb and has a DNA G + C content of 61.8 mol %. The GenBank/EMBL/DDBJ accession numbers for sequences of strain NC18^T^ generated in this study are as follows: MT367774 (16S rRNA gene), CP054858 (chromosome), CP054859 (plasmid), and CP054860 (plasmid).

## Figures and Tables

**Figure 1 microorganisms-09-01736-f001:**
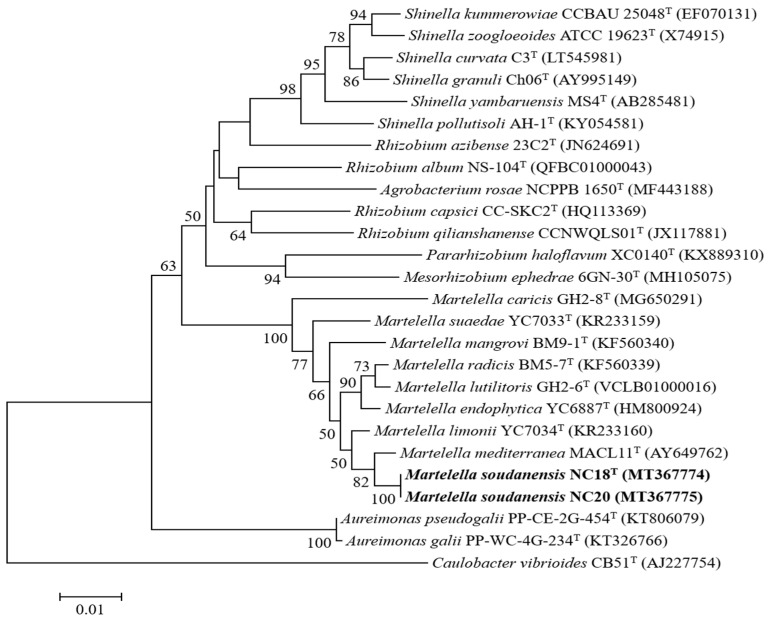
Phylogenetic tree based on 16S rRNA gene sequences of strains NC18^T^ and NC20 with other related taxa using 1393 bp sequence. Evolutionary distances calculated using the Jukes–Cantor model. Evolutionary history was inferred using the neighbor-joining method. Bootstrap values (tested as 1000 replications) above 50% are shown next to branches. Sequence of *Caulobacter vibrioides* CB51^T^ used as outgroup. Bar, 0.01 nucleotide substitution per position.

**Figure 2 microorganisms-09-01736-f002:**
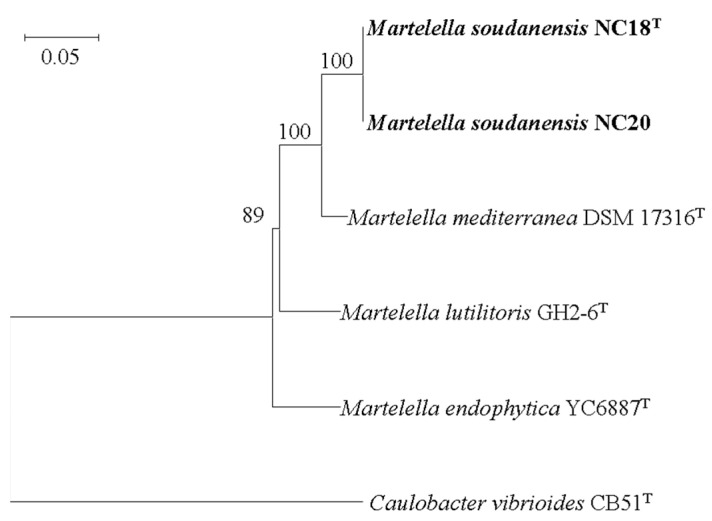
Multilocus sequence analysis (MLSA) tree based on 31 universally conserved protein sequences of strains NC18^T^ and NC20 with other related taxa. Evolutionary distances computed using JTT matrix-based method. Evolutionary history inferred using the neighbor-joining method. Bootstrap values (tested as 1000 replications) above 50% are shown next to branches. Sequence of *Caulobacter vibrioides* CB51^T^ used as outgroup. Bar, 0.05 nucleotide substitution per position. Accession numbers for each sequence are shown in Appendix A.

**Figure 3 microorganisms-09-01736-f003:**
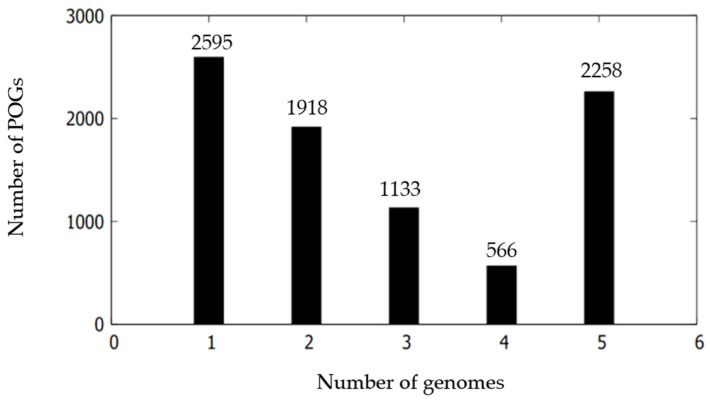
Comparison of POG distribution of strains NC18^T^ and NC20 with related taxa within the genus *Martelella*.

**Table 1 microorganisms-09-01736-t001:** Differential characteristics of strains NC18^T^ and NC20 with related taxa within the genus *Martelella*.

Characteristic	1	2	3	4	5	6	7	8	9	10
Cell size (μm)	0.8–0.9 × 1.2–1.3	0.6–0.8 × 1.1–1.3	1.0–1.1 × 1.3–1.5	0.8–1.1 × 1.2–2.5	0.5–0.8 × 1.2–2.3	1.0 × 1.2	0.8–0.9 × 1.9–2.8	0.4–0.7 × 1.0–1.5	1.0–1.2 × 1.2–1.5	0.8–0.9 × 1.3–1.4
Conditions for growth:										
Temperature (°C)	10–40 (30)	4–35 (30)	4–37 (28)	4–40 (25–30)	4–40 (28–30)	10–40	20–45 (30)	10–40 (25–30)	10–40	10–30 (30)
pH	5–9 (7)	5–9 (8)	5–8 (7)	5–9 (6–8.5)	5–9 (7–8.5)	5–9.5	5–10 (6–8)	5–10 (7–8)	5–8	4–10 (7–8)
NaCl (%)	0–13 (1)	0–13 (1)	0–5 (2)	1–6 (2–5)	0–9 (4–5)	2–12	0.5–9 (0.5–3)	1–9 (2–6)	2–10	0–11 (0–2)
Nitrate reduction	+	+	+	NR	+	-	-	NR	+	-
Aesculin hydrolysis	-	-	+	+	+	+	w	+	+	+
acetoin production	-	-	+	+	+	NR	NR	+	NR	NR
Fermentation-oxidation:										
L-rhamnose	+	+	-	-	-	NR	NR	-	NR	NR
D-glucose	-	-	NR	-	-	+	+	-	+	NR
D-mannitol	-	-	NR	-	-	NR	+	-	NR	NR
L-arabinose	+	+	NR	-	+	NR	+	+	NR	NR
Assimilation:										
potassium gluconate	-	-	+	-	-	-	-	+	-	-
malate	-	-	+	-	-	+	-	+	-	-
D-maltose	-	-	+	-	+	-	-	+	+	-
D-mannose	-	-	-	-	+	NR	-	+	NR	-
phenylacetate	-	-	-	NR	-	+	-	NR	+	-
D-glucose	-	-	+	-	+	+	-	+	+	-
D-mannitol	-	-	-	-	+	+	-	+	+	-
L-arabinose	-	-	-	-	+	NR	-	+	w	-
Enzyme activity:										
Urease	+	+	+	-	-	-	-	-	-	+
esterase lipase (C8)	-	-	+	+	-	+	+	+	+	+
lipase (C14)	-	-	+	-	-	-	-	-	-	-
trypsin	-	-	+	+	-	-	-	+	-	-
naphthol-AS-BI-phosphohydrolase	+	+	-	w	-	w	+	-	+	+
α-galactosidase	-	-	+	-	-	w	-	w	-	-
β-glucuronidase	-	-	+	-	-	-	-	+	-	-
β-glucosidase	+	-	+	-	+	+	+	+	+	+
N-acetyl-β-glucosaminidase	+	-	+	-	+	-	-	-	-	+
α-fucosidase	-	-	-	-	+	-	-	-	-	-
G + C content (%)	61.8	61.8	57.4	62.2	62.1	59.7	61.9	52.8	61	53.4

Strain: 1, NC18^T^; 2, NC20; 3, *M. mediterranea* DSM 17316^T^; 4, *M. limonii* YC7034^T^; 5, *M. endophytica* YC6887^T^; 6, *M. mangrovi* BM9-1^T^; 7, *M. lutilitoris* GH2-6^T^; 8, *M. suaedae* YC7033^T^; 9, *M. radicis* BM5-7^T^; 10, *M. caricis* GH2-8^T^. Data for strain 1 from this study. Data for strain 3 from Rivas et al. [1], strains 4 and 8 from Chung et al. [4], strain 5 from Bibi et al. [2], strains 6 and 9 from Zhang et al. [3], strain 7 from Kim et al. [6], and strain 10 from Lee [5]. +, positive; -, negative; w, weak; NR, not reported.

**Table 2 microorganisms-09-01736-t002:** Cellular fatty acid compositions (%) of strains NC18^T^ and NC20 with related taxa within the genus *Martelella*.

Fatty Acid	1	2	3	4	5	6	7	8	9	10
Saturated:										
C_16:0_	13.0	13.4	12.0	8.4	4.8	5.9	10.0	7.8	2.5	17.2
C_17:0_	0.8	0.7	-	1.1	-	1.2	<1	1.2	-	-
C_18:0_	6.2	6.3	4.3	6.9	8.8	8.9	10.4	6.6	7.6	2.9
C_19:0_ cyclo ω8c	25.8	28.2	41.4	25.3	28.0	23.7	9.8	20.8	24.9	12.5
Branched:										
C_18:1_ ω7c	-	-	21.7	-	17.9	35.7	-	-	41.7	-
C_18:1_ ω5c	0.4	0.4	-	-	-	-	-	-	-	-
C_18:1_ ω7c 11-methyl	6.4	6.6	8.8	17.8	7.9	5.4	8.4	16.2	6.8	8.9
C_19:0_ iso	0.3	-	-	-	-	-	-	-	-	-
Unsaturated:										
C_20:1_ ω7c	0.3	0.3	-	-	-	-	-	-	-	-
C_20:2_ ω6,9c	0.6	0.7	0.7	-	-	-	-	-	-	1.4
Hydroxy:										
C_16:0_ 3-OH	0.7	0.7	0.5	0.3	-	-	<1	0.3	-	0.9
C_18:1_ 2-OH	0.2	-	-	0.5	-	1.5	<1	0.5	1.9	0.9
C_18:0_ 3-OH	1.0	1.0	0.7	0.5	<1	0.7	<1	0.4	0.7	0.5
Summed feature*										
2	5.9	5.6	7.7	14.6	9.6	14.8	7.5	14.5	12.4	8.8
3	0.5	0.5	0.6	0.5	-	-	1.3	0.5	-	1.9
7	0.5	0.6	-	-	-	-	-	0.5	-	2.5
8	37.5	35.0	-	23.4	-	-	35.9	30.2	-	39.7

Strain: 1, NC18^T^; 2, NC20; 3, *M. mediterranea* DSM 17316^T^; 4, *M. limonii* YC7034^T^; 5, *M. endophytica* YC6887^T^; 6, *M. mangrovi* BM9-1^T^; 7, *M. lutilitoris* GH2-6^T^; 8, *M. suaedae* YC7033^T^; 9, *M. radicis* BM5-7^T^; 10, *M. caricis* GH2-8^T^. Data for strain 1 from this study. Data for strain 3 from Rivas et al. [1], strains 4 and 8 from Chung et al. [4], strain 5 from Bibi et al. [2], strains 6 and 9 from Zhang et al. [3], strain 7 from Kim et al. [6], and strain 10 from Lee [5]. Strain 1 was grown at MA for 2 days at 30 °C. -, not detected. *Summed features represent groups of two or three fatty acids that could not be separated by GLC with the MIDI system. Summed feature 2 contained C_12:0_ aldehyde and/or iso-C_16:1_ I and/or C_14:0_ 3-OH. Summed feature 3 contained C_16:1_ ω7c and/or C_16:1_ ω6c. Summed feature 7 contained unidentified 18.846 and/or C_19:1_ ω6c and/or C_19:0_ cyclo ω10c and/or C_19_ ω6. Summed feature 8 contained C_18:1_ ω7c and/or C_18:1_ ω6c.

**Table 3 microorganisms-09-01736-t003:** Genomic features of strains NC18^T^ and NC20 with related taxa within the genus *Martelella*.

Feature	1	2	3	4	5
Genome size (bp)	6,109,459	6,109,677	5,693,067	4,817,335	4,447,501
N50 (bp)	5,586,623	5,586,823	4,671,477	4,817,335	402,005
Contig number	3	3	4	1	18
Assembly level	complete	complete	complete	complete	contig
Total genes	5849	5830	5303	4539	4093
Protein coding genes	5502	5585	5115	4409	3981
rRNA	6	6	6	9	3
tRNA	48	48	48	53	47
GenBank assembly accession number	GCA_013459415.1	GCA_013459645.1	GCA_002043005.1	GCA_000960975.1	GCA_005924265.1

Strain: 1, NC18^T^; 2, NC20; 3, *M. mediterranea* DSM 17316^T^; 4, *M. endophytica* YC6887^T^; 5, *M. lutilitoris* GH2-6^T^.

**Table 4 microorganisms-09-01736-t004:** ANI values (%) and dDDH values (%) of strains NC18^T^ and NC20 with related taxa within the genus *Martelella*.

Strain	ANI Values (%)		dDDH Values (%)	
	NC18^T^	NC20	NC18^T^	NC20
*Martelella soudanensis* NC18^T^	-	99.9	-	100
*Martelella soudanensis* NC20	99.9	-	100	-
*Martelella mediterranea* DSM 17316^T^	88.1	88.0	34.9	34.9
*Martelella endophytica* YC6887^T^	80.2	80.4	23.9	23.9
*Martelella lutilitoris* GH2-6^T^	80.4	80.4	23.7	23.7

**Table 5 microorganisms-09-01736-t005:** AAI results of strains NC18^T^ and NC20 with related taxa within the genus *Martelella*.

Strains	*M. soudanensis* NC18^T^	*M. soudanensis* NC20	*M. mediterranea* DSM 17316^T^	*M. endophytica* YC6887^T^	*M. lutilitoris* GH2-6^T^
*M. soudanensis* NC18^T^		100	87.7	76.3	77.8
*M. soudanensis* NC20	100		87.9	76.5	78.0
*M. mediterranea* DSM 17316^T^	87.7	87.9		77.3	78.2
*M. endophytica* YC6887^T^	76.3	76.5	77.3		78.5
*M. lutilitoris* GH2-6^T^	77.8	78.0	78.2	78.5	

## Data Availability

Generated sequences can be found as stated under the species description.

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
