# Peer review of "Characterization of *Martelella soudanensis* sp. nov., Isolated from a Mine Sediment"

_microorganisms, 2021, doi:10.3390/microorganisms9081736_

Round 1
Reviewer 1 Report
Jung-Yun et al., describe two novel type strains of Martelella soudanesis, NC18T and NC20, and fully characterize their phenotypic and metabolic traits. Highlighted from these novel strains is an ability to grow in increased saline, different pH and temperatures, and other enzymatic activity different from the other species of this genus. The manuscript is extremely thorough and has great details in the experimental methods used and the data depostision and acquisition information. Characterizing novel species capable of variable growth is critical for expanding biotechnological applications and utilizing bacteria to their greatest power for society. With the great data presentation and explanations in this manuscript, I recommend for its publication with no revisions. One typo was noticed:
Line 129: change “maxi-mum-likelihood” to “maximum-likelihood”
Reviewer 2 Report
The paper is a thorough study on two halophilic isolates from the genus Martelella isolates from the sediment of a 714 m subsurface borehole in the Soudan Mine in Minnesota. The combination of polyphasic taxonomy techniques with pan-genomic analysis provides evidence for the erection of a new species M. soudanensis. The paper is well and clearly written and, by addressing a problem related with microbial diversity, is well within the scope of the Environmental Microbiology section of Microorganisms. I recommend publication in the present form.